# Symmetries in quantum networks lead to no-go theorems for entanglement distribution and to verification techniques

Kiara Hansenne [1], Zhen-Peng Xu [1✉], Tristan Kraft[1,2] & Otfried Gühne [1]

Quantum networks are promising tools for the implementation of long-range quantum communication. The characterization of quantum correlations in networks and their usefulness for information processing is therefore central for the progress of the field, but so far only results for small basic network structures or pure quantum states are known. Here we show that symmetries provide a versatile tool for the analysis of correlations in quantum networks. We provide an analytical approach to characterize correlations in large network structures with arbitrary topologies. As examples, we show that entangled quantum states with a bosonic or fermionic symmetry can not be generated in networks; moreover, cluster and graph states are not accessible. Our methods can be used to design certification methods for the functionality of specific links in a network and have implications for the design of future network structures.

[1] Naturwissenschaftlich-Technische Fakultät, Universität Siegen, Siegen, Germany. [2]Present address: Institute for Theoretical Physics, University of Innsbruck, Innsbruck, Austria. ✉email: zhen-peng.xu@uni-siegen.de

A central paradigm for quantum information processing is the notion of quantum networks[1–4]. In an abstract sense, a quantum network consists of quantum systems as nodes on specific locations, where some of the nodes are connected via links. These links correspond to quantum channels, which may be used to send quantum information (e.g., a polarized photon) or where entanglement may be distributed. Crucial building blocks for the links, such as photonic quantum channels between a satellite and ground stations[5–8] or the high-rate distribution of entanglement between nodes[9,10] have recently been experimentally demonstrated. Clearly, such real physical implementations are always noisy and may only work probabilistically, but there are various theoretical approaches to deal with this[11–14].

For the further progress of the field, it is essential to design methods for the certification and benchmarking of a given network structure or a single specific link within it. In view of current experimental limitations, the question arises which states can be prepared in the network with moderate effort, e.g., with simple local operations. This question has attracted some attention, with several lines of research emerging. First, the problem has been considered in the classical setting, such as the analysis of causal structures[15–17] or in the study of hidden variable models, where the hidden variables are not equally distributed between every party [18–24]. Concerning quantum correlations, several initial works appeared in the last year, suggesting slightly different definitions of network entanglement[25–28]. These have been further investigated [29–31] and methods from the classical realm have been extended to the quantum scenario[32]. Still, the present results are limited to simple networks like the triangle network, noise-free networks or networks build from specific quantum states, or bounded to small dimensions due to numerical limitations.

In this work, we show an analytical approach to characterize quantum correlations in arbitrary network topologies. Our approach is based on symmetries, which may occur as permutation symmetries or invariance under certain local unitary transformations. Symmetries play an outstanding role in various fields of physics[33] and they have already turned out to be useful for various other problems in quantum information theory[34–41]. On a technical level, we combine the inflation technique for quantum networks[17,27] with estimates known from the study of entropic uncertainty relations[42–44]. Based on our approach, we derive simply testable inequalities in terms of expectation values, which can be used to decide whether a given state may be prepared in a network or not. With this we can prove that large classes of states cannot be prepared in networks using simple communication, for instance all multiparticle graph states with up to twelve vertices with noise, as well as all mixed entangled permutationally symmetric states. This delivers various methods for benchmarking: First, the observation of such states in a network certifies the implementation of advanced network protocols. Second, our results allow to design simple tests for the proper working of a specific link in a given network.

## Results

**Network entanglement.** To start, let us define the types of correlations that can be prepared in a network. In the simplest scenario Alice, Bob and Charlie aim to prepare a tripartite quantum state using three bipartite source states $\varrho_a$, $\varrho_b$ and $\varrho_c$, see Fig. 1. Parties belonging to a same source state are sent to different parties of the network, i.e., $A$, $B$ or $C$, such that the global state reads $\varrho_{ABC} = \varrho_c \otimes \varrho_b \otimes \varrho_a$. Note that here the order of the parties on both sides of the equation is different. After receiving the states, each party may still apply a local operation $\mathcal{E}_X$ (for $X = A, B, C$), in addition these operations may be coordinated by

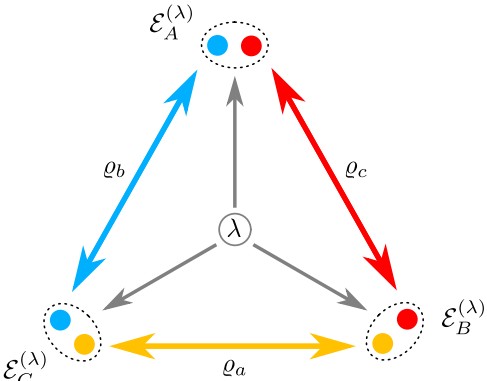

**Fig. 1 Triangle quantum network.** Three sources $\varrho_a$, $\varrho_b$ and $\varrho_c$ distribute parties to three nodes, Alice, Bob, and Charlie ($A$, $B$, and $C$). The colors yellow, blue, and red are associated to the sources $\varrho_a$, $\varrho_b$, and $\varrho_c$ respectively. Alice, Bob, and Charlie each end up with a bipartite system $X = X_1 X_2$ on which they perform a local channel $\mathcal{E}_X^{(\lambda)}$ ($X = A, B, C$) depending on a classical random variable $\lambda$.

shared randomness. This leads to a global state of the form

$$\varrho = \sum_\lambda p_\lambda \mathcal{E}_A^{(\lambda)} \otimes \mathcal{E}_B^{(\lambda)} \otimes \mathcal{E}_C^{(\lambda)} [\varrho_{ABC}], \tag{1}$$

and the question arises, which three-party states can be written in this form and which cannot?

Some remarks are in order: First, the definition of network states in Eq. (1) can directly be extended to more parties or more advanced sources, e.g., one can consider the case of five parties $A, B, C, D, E$, where some sources distribute four-party states between some of the parties. Second, the scenario considered uses local operations and shared randomness (LOSR) as allowed operations, which is a smaller set than local operations and classical communication (LOCC). In fact, LOCC are much more difficult to implement, but using LOCC and teleportation any tripartite state can be prepared from bipartite sources. On the other hand, the set LOSR is strictly larger than, e.g., the unitary operations considered in refs. [25,26]. Finally, the discerning reader may have noticed that in Eq. (1) the state $\varrho_{ABC}$ does not depend on the shared random variable $\lambda$, but since the dimension of the source states $\varrho_i$ is not bounded one can always remove a dependency on $\lambda$ in the $\varrho_i$ by enlarging the dimension[27]. Equivalently, one may remove the dependency of the maps $\mathcal{E}_X$ on $\lambda$ and the shared randomness may be carried by the source states only.

**Symmetries.** Symmetry groups can act on quantum states in different ways. First, the elements of a unitary symmetry group may act transitively on the density matrix $\varrho$. That is, $\varrho$ is invariant under transformations like

$$\varrho \longmapsto U \varrho U^\dagger = \varrho. \tag{2}$$

If $\varrho = |\psi\rangle\langle\psi|$ is pure, this implies $U|\psi\rangle = e^{i\phi}|\psi\rangle$ and $|\psi\rangle$ is, up to some phase, an eigenstate of some operator. Second, for pure states one can also identify directly a certain subspace of the entire Hilbert space that is equipped with a certain symmetry, e.g., symmetry under exchange of two particles. Denoting by $\Pi$ the projector onto this subspace, the symmetric pure states are defined via

$$\Pi|\psi\rangle = |\psi\rangle \tag{3}$$

and for mixed states one has $\varrho = \Pi \varrho \Pi$. Note that if $\varrho = \sum_k p_k |\phi_k\rangle\langle\phi_k|$ has some decomposition into pure states, then each $|\phi_k\rangle = \Pi|\phi_k\rangle$ has to be symmetric, too.

In the following, we consider mainly two types of symmetries. First, we consider multi-qubit states obeying a symmetry as in Eq. (2) where the symmetry operations consist of an Abelian group of tensor products of Pauli matrices. These groups are usually referred to as stabilizers in quantum information theory[45], and they play a central role in the construction of quantum error correcting codes. Pure states obeying such symmetries are also called stabilizer states, or, equivalently, graph states[46,47]. Second, we consider states with a permutational (or bosonic) symmetry[36,39,48], obeying relations as in Eq. (3) with $\Pi$ being the projector onto the symmetric subspace.

**GHZ states.** As a warming-up exercise we discuss the Greenberger-Horne-Zeilinger (GHZ) state of three qubits,

$$|GHZ\rangle = \frac{1}{\sqrt{2}}(|000\rangle + |111\rangle) \tag{4}$$

in the triangle scenario. This simple case was already the main example in previous works on network correlations[25–27], but it allows us to introduce our concepts and ideas in a simple setting, such that their full generalization is later conceivable.

The GHZ state is an eigenstate of the observables

$$g_1 = X_A X_B X_C, \quad g_2 = 1_A Z_B Z_C, \quad g_3 = Z_A Z_B 1_C. \tag{5}$$

Here and in the following we use the shorthand notation $1_A X_B Y_C = \mathbb{1} \otimes \sigma_x \otimes \sigma_y$ for tensor products of Pauli matrices. Indeed, these $g_k$ commute and generate the stabilizer $\mathcal{S} = \{\mathbb{1}, g_1, g_2, g_3, g_1g_2, g_1g_3, g_2g_3, g_1g_2g_3\}$. Clearly, for any $S_i \in \mathcal{S}$ we have $S_i|GHZ\rangle = |GHZ\rangle$ and so $\langle GHZ|S_i|GHZ\rangle = 1$.

As a tool for studying network entanglement, we use the inflation technique[17,32]. The basic idea is depicted in Fig. 2. If a state $\varrho$ can be prepared in the network scenario, then one can also consider a scenario where each source state is sent two-times to multiple copies of the parties. In this multicopy scenario, the source states may, however, also be wired in a different manner. In the simplest case of doubled sources, this may lead to two different states, $\tau$ and $\gamma$. Although $\varrho$, $\tau$ and $\gamma$ are different states, some of their marginals are identical, see Fig. 2 and Supplementary Note 1. If one can prove that states $\tau$ and $\gamma$ with the marginal conditions do not exist, then $\varrho$ cannot be prepared in the network.

Let us start by considering the correlation $\langle Z_A Z_B \rangle$ in $\varrho$, $\tau$ and $\gamma$. The values are equal in all three states,

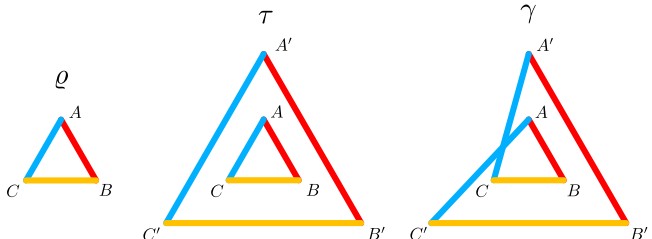

**Fig. 2 Triangle network and two of its inflations.** The first figure represents the triangle network of Fig. 1, with global state $\varrho$ and parties A, B and C. Using the same source states (represented by lines of same color, i.e., yellow, blue and red are associated to the sources $\varrho_a$, $\varrho_b$ and $\varrho_c$ of Fig. 1 respectively) and same local channels, one can build the so-called inflated state $\tau$ with parties X, X' (X = A, B, C). This state is separable with respect to the ABC|A'B'C' partition. The state $\gamma$ is build similarly, but with a rewiring of the sources, leading to an inflated state that is in general not separable and different from $\tau$. Still, this procedure imposes that several marginals of $\varrho$, $\tau$ and $\gamma$ are equal, e.g., $\varrho_{AC} = \tau_{A'C'} = \gamma_{A'C}$. The parties of $\gamma$ are labeled in the same way than $\tau$. Note that this is a simplified version of Fig. 1, i.e., that the local channels and the randomness source are not depicted but implied.

$\langle Z_A Z_B \rangle_{\varrho} = \langle Z_A Z_B \rangle_{\tau} = \langle Z_A Z_B \rangle_{\gamma}$, and the same holds for the correlation $\langle Z_B Z_C \rangle$. Note that these should be large, if $\varrho$ is close to a GHZ state, as $Z_A Z_B$ is an element of the stabilizer. Using the general relation $\langle Z_A Z_C \rangle \geq \langle Z_A Z_B \rangle + \langle Z_B Z_C \rangle - 1$[27] we can use this to estimate $\langle Z_A Z_C \rangle$ in $\gamma$. Due to the marginal conditions, we have $\langle Z_A Z_C \rangle_{\gamma} = \langle Z_{A'} Z_C \rangle_{\tau}$, implying that this correlation in $\tau$ must be large, if $\varrho$ is close to a GHZ state. On the other hand, the correlation $\langle X_A X_B X_C \rangle$ corresponds also to a stabilizer element and should be large in the state $\varrho$ as well as in $\tau$.

The key observation is that the observables $X_A X_B X_C$ and $Z_{A'} Z_C$ anticommute; moreover, they have only eigenvalues $\pm 1$. For this situation, strong constraints on the expectation values are known: If $M_i$ are pairwise anticommuting observables with eigenvalues $\pm 1$, then $\sum_i \langle M_i \rangle^2 \leq 1$[44]. This fact has already been used to derive entropic uncertainty relations[42,43] or monogamy relations[49,50]. For our situation, it directly implies that for $\tau$ the correlations $\langle X_A X_B X_C \rangle$ and $\langle Z_{A'} Z_C \rangle$ cannot both be large. Or, expressing everything in terms of the original state $\varrho$, if $\langle Z_A Z_B \rangle + \langle Z_B Z_C \rangle - 1 \geq 0$ then a condition for preparability of a state in the network is

$$\langle X_A X_B X_C \rangle^2 + \left( \langle Z_A Z_B \rangle + \langle Z_B Z_C \rangle - 1 \right)^2 \leq 1. \tag{6}$$

This is clearly violated by the GHZ state. In fact, if one considers a GHZ state mixed with white noise, $\varrho = p|GHZ\rangle\langle GHZ| + (1-p)\mathbb{1}/8$, then these states are detected already for $p > 4/5$. Note that using the other observables of the stabilizer and permutations of the particles, also other conditions like $\langle Y_A Y_B X_C \rangle^2 + \left( \langle Z_A Z_B \rangle + \langle Z_A Z_C \rangle - 1 \right)^2 \leq 1$ can be derived.

Using these techniques as well as concepts based on covariance matrices[28,29] and classical networks[23], one can also derive bounds on the maximal GHZ fidelity achievable by network states. In fact, for network states

$$F_{GHZ} = \langle GHZ|\varrho|GHZ\rangle \leq \frac{1}{\sqrt{2}} \approx 0.7071 \tag{7}$$

holds, as explained in Supplementary Note 2. This is a clear improvement on previous analytical bounds, although it does not improve a fidelity bound obtained by numerical convex optimization[27].

**Cluster and graph states.** The core advantage of our approach is the fact that it can directly be generalized to more parties and complicated networks, while the existing numerical and analytical approaches are mostly restricted to the triangle scenario.

Let us start the discussion with the four-qubit cluster state $|C_4\rangle$. This may be defined as the unique common +1-eigenstate of

$$g_1 = X_A Z_B 1_C Z_D, \quad g_2 = Z_A X_B Z_C 1_D,$$
$$g_3 = 1_A Z_B X_C Z_D, \quad g_4 = Z_A 1_B Z_C X_D. \tag{8}$$

For later generalization, it is useful to note that the choice of these observables is motivated by a graphical analogy. For the square graph in Fig. 3 one can associate to any vertex a stabilizing operator in the following manner: One takes $X$ on the vertex $i$, and $Z$ on its neighbors, i.e., the vertices connected to $i$. This delivers the observables in Eq. (8), but it may also be applied to general graphs, leading to the notion of graph states[46]. 

If a quantum state can be prepared in the square network, then we can consider the third order inflated state $\xi$ shown in Fig. 3. In the inflation $\xi$, the three observables $X_{B''}X_D$, $Z_{B'}X_C Z_D$, and $X_A Y_B Y_D$ anticommute. These observables act on marginals that are identical to those in $\varrho$. Consequently, for any state that can be

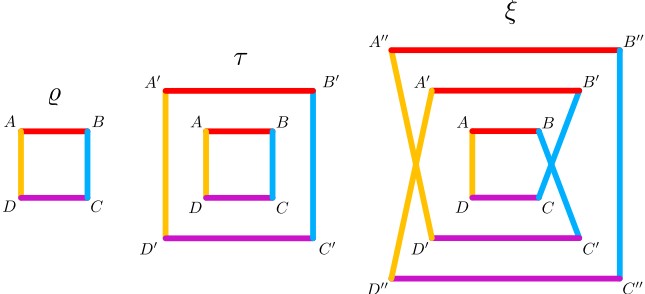

**Fig. 3 Square network and two of its inflations.** Similar to the triangle network of Fig. 2, the state $\tau$ is generated using two copies of the sources and channels used to generate $\varrho$. Then, one goes to a higher-order inflation by using three copies of the sources and the local channels. By rewiring one obtains the inflated state $\xi$. Again, one has several equalities between the marginals of $\varrho$, $\tau$, and $\xi$. Parties with a link of same color are connected by an identical source, as in Fig. 2. The labels $X$, $X'$, $X''$ ($X = A, B, C$) denote the parties of the states.

prepared in the square network, the relation

$$\langle X_B X_D \rangle^2 + \langle Z_B X_C Z_D \rangle^2 + \langle X_A Y_B Y_D \rangle^2 \leq 1 \qquad (9)$$

holds. All these observables are also within the stabilizer of the cluster state, so the cluster state violates this inequality with an lhs equal to three. This proves that cluster states mixed with white noise cannot be prepared in the square network for $p > 1/\sqrt{3} \approx 0.577$. Again, with the same strategy different nonlinear witnesses like $\langle X_A X_C \rangle^2 + \langle Y_A Y_B Z_C Z_D \rangle^2 \leq 1$ for network entanglement in the square network can be derived. This follows from the second inflation in Fig. 3. Furthermore, it can be shown that states with a cluster state fidelity of $F_{C_4} > 0.7377$ cannot be prepared in a network. The full discussion is given in Supplementary Note 3.

This approach can be generalized to graph states. As already mentioned, starting from a general graph one can define a graph state by stabilizing operators in analogy to Eq. (8). The resulting states play an eminent role in quantum information processing. For instance, the so-called cluster states, which correspond to graphs of quadratic and cubic square lattices, are resource states for measurement-based quantum computation[51] and topological error correction[52,53].

Applying the presented ideas to general graphs results in the following: If a graph contains a triangle, then under simple and weak conditions an inequality similar to Eq. (6) can be derived. This then excludes the preparability of noisy graph states in any network with bipartite sources only. Note that this is a stronger statement than proving network entanglement only for the network corresponding to the graph, as was considered above for the cluster state.

At first sight, the identification of a specific triangle in the graph may seem a weak condition, but here the entanglement theory of graph states helps: It is well known that certain transformations of the graph, so-called local-complementations, change the graph state only by a local unitary transformation[54,55], so one may apply these to generate the triangle with the required properties. Indeed, this works for all cases we considered (e.g., the full classification up to twelve qubits from refs. [56–58]) and we can summarize:

*Observation 1.* (a) No graph state with up to twelve vertices can be prepared in a network with only bipartite sources. (b) If a graph contains a vertex with degree $d \leq 3$, then it cannot be prepared in any network with bipartite sources. (c) The two- and three-dimensional cluster states cannot be prepared in any network.

In all the cases, it follows that graph states mixed with white noise, $\varrho = p|G\rangle\langle G| + (1-p)\mathbb{1}/2^N$ are network entangled for $p > 4/5$, independently of the number of qubits. A detailed discussion is given in Supplementary Note 3.

As mentioned above, the exclusion of noisy graph states from the set of network states with bipartite sources holds for all graph states we considered. Therefore, we conjecture that this is valid for all graph states, without restrictions on the number of parties.

We note that similar statements on entangled multiparticle states and symmetric states were made in ref. [26]. However, we stress that the methods to obtain these results are very different from the anticommuting method used here, and that Observation 1 is only an application of this method (see section on the certification of network links for another use). Furthermore, the result of ref. [26] concerning permutationally symmetric states only holds for pure states, whereas in the next section we will see that it holds for all permutationally symmetric states.

A natural questions that arises is whether this method might be useful to characterize correlations in networks with more-than-bipartite sources. While this still needs to be investigated in details, examples show that the answer is most likely positive: Using the anticommuting relations, we demonstrate in Supplementary Note 3 that some states cannot be generated in networks with tripartite sources.

**Permutational symmetry**. Now we consider multiparticle quantum states of arbitrary dimension that obey a permutational or bosonic symmetry. Mathematically, these states act on the symmetric subspace only, meaning that $\varrho = \Pi^+ \varrho \Pi^+$, where $\Pi^+$ is the projector on the symmetric subspace. For example, in the case of three qubits this space is four-dimensional, and spanned by the Dicke states $|D_0\rangle = |000\rangle$, $|D_1\rangle = (|001\rangle + |010\rangle + |100\rangle)/\sqrt{3}$, $|D_2\rangle = (|011\rangle + |101\rangle + |110\rangle)/\sqrt{3}$, and $|D_3\rangle = |111\rangle$.

The symmetry has several consequences[36,39]. First, if one has a decomposition $\varrho = \sum_k p_k |\psi_k\rangle\langle\psi_k|$ into pure states, then all $|\psi_k\rangle$ have to come from the symmetric space too. Since pure symmetric states are either fully separable (like $|D_0\rangle$) or genuine multiparticle entangled (like $|D_1\rangle$), this implies that mixed symmetric states have also only these two possibilities. That is, if a mixed symmetric state is separable for one bipartition, it must be fully separable.

Second, permutational invariance can also be characterized by the flip operator $F_{XY} = \sum_{ij} |ij\rangle_{XY}\langle ji|_{XY}$ on the particles $X$ and $Y$. Symmetric multiparticle states obey $\varrho = F_{XY}\varrho = \varrho F_{XY}$ for any pair of particles, where the second equality directly follows from hermiticity. Conversely, concluding full permutational symmetry from two-particle properties only requires this relation for pairs such that the $F_{XY}$ generate the full permutation group. Finally, it is easy to check that if the marginal $\varrho_{XY}$ of a multiparticle state $\varrho$ obeys $\varrho_{XY} = F_{XY}\varrho_{XY}$, then the full state $\varrho$ obeys the same relation, too.

Armed with these insights, we can explain the idea for our main result. Consider a three-particle state with bosonic symmetry that can be prepared in a triangle network, the inflation $\gamma$ from Fig. 2 and the reduced state $\gamma_{ABC}$ in this inflation. This obeys $\gamma_{ABC} = F_{XY}\gamma_{ABC}$ for $XY$ equal to $AB$ or $BC$. Since $F_{AB}F_{BC}F_{AB} = F_{AC}$, this implies that the reduced state $\tau_{AC'}$ obeys $\tau_{AC'} = F_{AC'}\tau_{AC'}$ and hence also the six-particle state $\tau$. Moreover, $\tau$ also obeys similar constraints for other pairs of particles (like $AB$, $BC$, $A'B'$ and $B'C'$) and it is easy to see that jointly with $AC'$ these generate the full permutation group. So, $\tau$ must be fully symmetric. But $\tau$ is separable with respect to the $ABC|A'B'C'$ bipartition, so $\tau$ and hence $\varrho = \tau_{ABC}$ must be fully separable.

The same argument can easily be extended to more complex networks, which are not restricted to use bipartite sources and holds for states of arbitrary local dimension. We can summarize:

*Observation 2*. Consider a permutationally symmetric state of $N$ parties. This state can be generated by a network with $(N-1)$-partite sources if and only if it is fully separable.

We add that this Observation can also be extended to the case of fermionic antisymmetry, a detailed discussion is given in the Supplementary Note 4.

**Certifying network links**. For the technological implementation of quantum networks, it is of utmost importance to design certification methods to test and benchmark different realizations. One of the basic questions is, whether a predefined quantum link works or not. Consider a network where the link between two particles is absent or not properly working. For definiteness, we may consider the square network on the lhs of Fig. 3 and the parties $A$ and $C$. In the second inflation $\tau$ we have for the marginals $\tau_{AC} = \tau_{A'C}$. This implies that the observables $X_A X_C$ and $Z_A Z_C$ on the original state $\varrho$ correspond to anticommuting observables on $\tau$, so we have $\langle X_A X_C \rangle^2 + \langle Z_A Z_C \rangle^2 \leq 1$. Using higher-order inflations, one can extend and formulate it for general networks: If a state can be prepared in a network with bipartite sources but without the link $AC$, then

$$\langle X_A X_C P_{R_1} \rangle^2 + \langle Y_A Y_C P_{R_2} \rangle^2 + \langle Z_A Z_C P_{R_3} \rangle^2 \leq 1. \tag{10}$$

Here the $P_{R_i}$ are arbitrary observables on disjoint subsets of the other particles, $R_i \cap R_j = \emptyset$. If a state was indeed prepared in a real quantum network then violation of this inequality proves that the link $AC$ is working and distributing entanglement. In Supplementary Note 5, details are discussed and examples are given, where this test allows to certify the functionality of a link even if the reduced state $\varrho_{AC}$ is separable.

## Discussion

We have provided an analytical method to analyze correlations arising in quantum networks from few measurements. With this, we have shown that large classes of states with symmetries, namely noisy graph states and permutationally symmetric states cannot be prepared in networks. Moreover, our approach allows to design simple tests for the functionality of a specified link in a network.

Our results open several research lines of interest. First, they are of direct use to analyze quantum correlations in experiments and to show that multiparticle entanglement is needed to generate observed quantum correlations. Second, they are useful for the design of networks in the realistic setting: For instance, we have shown that the generation of graph states from bipartite sources necessarily requires at least some communication between the parties, which may be of relevance for quantum repeater schemes based on graph states that have been designed[59]. Moreover, it has been shown that GHZ states provide an advantage for multipartite conference key agreement over bipartite sources[60], which may be directly connected to the fact that their symmetric entanglement is inaccessible in networks. Finally, our results open the door for further studies of entanglement in networks, e.g., using limited communication (first results on this have recently been reported[61]) or restricted quantum memories, which is central for future realizations of a quantum internet.

## Data availability

Data sharing not applicable to this article as no data sets were generated or analyzed during the current study.

## Code availability

The codes used for this study are available from the corresponding author upon reasonable request.

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

## Acknowledgements

We thank Xiao-Dong Yu and Carlos de Gois for discussions. This work was supported by the Deutsche Forschungsgemeinschaft (DFG, German Research Foundation, project numbers 447948357 and 440958198), the Sino-German Center for Research Promotion (Project M-0294), and the ERC (Consolidator Grant 683107/TempoQ). K.H. acknowledges support from the House of Young Talents of the University of Siegen. Z.P.X. acknowledges support from the Humboldt foundation. T.K. acknowledges support from the Austrian Science Fund (FWF): P 32273-N27.

## Author contributions

K.H., Z.P.X., T.K. and O.G. derived the results and wrote the manuscript. K.H. and Z.P.X. contributed equally to the project. O.G. supervised the project.

## Funding

## Competing interests

The authors declare no competing interests.
