## [Peer Review File · Nature Communications]

Symmetries in quantum networks lead to no-go theorems for entanglement distribution and to verification techniquesReviewers' Comments:

Reviewer #1:

Remarks to the Author:

The submitted manuscript deals with a very timely topic, the characterization of quantum correlations that can be observed in networks. Namely, the network geometry severely restricts types of quantum correlations. The authors focus on density matrices, i.e. quantum states, and try to answer how we can determine what kind of quantum states can be generated with a given network. Up to now the main method for approaching this problem was using inflation techniques. However, the technique is applicable only for networks involving a rather small number of nodes and edges. In this manuscript the authors suggest using symmetry to make the use of inflation more economical. They show that the method works very well for stabilizer states and those with bosonic or fermionic symmetry.

I like the problem the authors consider. The method might be quite useful in further research, and I think that this paper should be highlighted by publication in Nature Communications. It considers a fairly unexplored problem and provides a concrete step forward. In addition, the paper is well written and easy to read and understand even by non-specialists.

Reviewer #2:

Remarks to the Author:

This manuscript is concerned with the quantum networks scenario which consists of many spatially separated observers that are linked by different sources of quantum correlations and can perform local operations on their shares of distributed quantum states. Moreover, the implementation of these local channels is coordinated by classical correlations distributed to the observers by a single source. On the other hand, the authors impose one constraint on the considered scenario that no classical communication between parties is allowed. This assumption is well motivated in the manuscript and at the same time makes the considered scenario highly nontrivial.

Quantum networks constitute a paradigmatic framework in quantum information as it models for instance quantum communication protocols or quantum internet. For this reason characterization of quantum correlations that can be prepared within quantum networks under various assumptions is gaining a lot of attention recently. At the same time this problem is extremely difficult to study both from analytical and numerical perspectives and therefore most of the results on distribution of entangled states concern networks of the simplest topologies such the triangle networks.

In the present manuscript, by employing certain symmetries that play a prominent role in quantum information or physics in general, which are the permutational symmetry and invariance under the action of certain stabilizer groups, the authors are able to go significantly beyond what has been known so far. Precisely, they provide two no-go theorems for distribution of entanglement in quantum networks that can be summarized as follows: (i) a certain large class of N -particle stabilizer states cannot be prepared in quantum networks with bipartite sources, (ii) no symmetric entangled mixed states can be prepared in networks with $(N-1)$ -partite sources. These statements are then used to design methods of certifying that a given link of the network functions properly.

I find this paper very interesting and important. By nicely combining certain symmetries with the inflation method it significantly moves our understanding of quantum networks forward. Consequently, I am happy to recommend it for publication in Nature Communications. There are a few suggestions and corrections that can be taken into account while preparing the final version:

1. It is not entirely clear, at least to me, whether Observation 2 concerns the symmetric states of arbitrary local dimension, or similarly to Observation 1 only the qubit ones. I think it would be useful

to reveal that information in the main text.

2. I don't quite understand why the authors cited [26] in a footnote. If Ref. [26] contains similar statements to those made in Observation 1 (as the authors state in the footnote), it would better to mention that explicitly in the text. In fact, the authors could add a sentence or two with a short comparison between the present results and those of [26].

3. I think it would be useful to add a few sentences commenting on how the results on the stabilizer states would change if the authors considered sources that link more parties than two.

4. In Supplementary Note there are some problems with references to Figures and Equations [between Eqs. (3) and (4) and Eqs. (40) and (41)].

5. There is redundant word 'Interestingly' in the third sentence of the section 'Certifying network links'.

6. The last sentence of the work: 'result open' → 'result opens'.

REPLY TO REVIEWERS' COMMENTS

Reviewer #1 (Remarks to the Author):

The submitted manuscript deals with a very timely topic, the characterization of quantum correlations that can be observed in networks. Namely, the network geometry severely restricts types of quantum correlations. The authors focus on density matrices, i.e. quantum states, and try to answer how we can determine what kind of quantum states can be generated with a given network. Up to now the main method for approaching this problem was using inflation techniques. However, the technique is applicable only for networks involving a rather small number of nodes and edges. In this manuscript the authors suggest using symmetry to make the use of inflation more economical. They show that the method works very well for stabilizer states and those with bosonic or fermionic symmetry.

I like the problem the authors consider. The method might be quite useful in further research, and I think that this paper should be highlighted by publication in Nature Communications. It considers a fairly unexplored problem and provides a concrete step forward. In addition, the paper is well written and easy to read and understand even by non-specialists.

Our reply: We would like to thank the reviewer for the time they took to read and comment on our manuscript. We are glad the reviewer liked our manuscript and recommended it for publication.

Reviewer #2 (Remarks to the Author):

This manuscript is concerned with the quantum networks scenario which consists of many spatially separated observers that are linked by different sources of quantum correlations and can perform local operations on their shares of distributed quantum states. Moreover, the implementation of these local channels is coordinated by classical correlations distributed to the observers by a single source. On the other hand, the authors impose one constraints on the considered scenario that no classical communication between parties is allowed. This assumption is well motivated in the manuscript and at the same time makes the considered scenario highly nontrivial.

Quantum networks constitute a paradigmatic framework in quantum information as it models for instance quantum communication protocols or quantum Internet. For this reason characterization of quantum correlations that can be prepared within quantum networks under various assumptions is gaining a lot of attention recently. At the same time this problem is extremely difficult to study both from analytical and numerical perspectives and therefore most of the results on distribution of entangled states concern networks of the simplest topologies such the triangle networks.

In the present manuscript, by employing certain symmetries that play a prominent role in quantum information or physics in general, which are the permutational symmetry and invariance under the action of certain stabilizer groups, the authors are able to go significantly beyond what has been known so far. Precisely, they provide two no-go theorems for distribution of entanglement in quantum networks that can be summarized as follows: (i) a certain large class of N -particle stabilizer states cannot be prepared in quantum networks with bipartite sources, (ii) no symmetric entangled mixed states can be prepared in networks with $(N-1)$ -partite sources. These statements are then used to design methods of certifying that a given link of the network functions properly.

I find this paper very interesting and important. By nicely combining certain symmetries with the

inflation method it significantly moves our understanding of quantum networks forward. Consequently, I am happy to recommend it for publication in Nature Communications. There are a few suggestions and corrections that can be taken into account while preparing the final version:

Our reply: We would like to thank the reviewer for the time they took to read and evaluate our paper. We are glad the reviewer found our manuscript interesting and recommended it for publication.

1. It is not entirely clear, at least to me, whether Observation 2 concerns the symmetric states of arbitrary local dimension, or similarly to Observation 1 only the qubit ones. I think it would be useful to reveal that information in the main text.

Our reply: We thank the reviewer for pointing this out, it indeed holds for arbitrary local dimensions. A sentence has been added above Observation 2 to clarify this point.

2. I don't quite understand why the authors cited [26] in a footnote. If Ref. [26] contains similar statements to those made in Observation 1 (as the authors state in the footnote), it would better to mention that explicitly in the text. In fact, the authors could add a sentence or two with a short comparison between the present results and those of [26].

Our reply: We thank the reviewer for addressing this. The footnote has been removed and a paragraph has been added to compare our results to the ones of Ref. [26].

3. I think it would be useful to add a few sentences commenting on how the results on the stabilizer states would change if the authors considered sources that link more parties than two.

Our reply: We thank the reviewers for this comment. We added a paragraph (page 4, above **Permutational symmetry**) commenting on this. It turns out that our method based on an anticommuting relation might be generalized to more than bipartite sources. We present in the supplemental material two different examples of the method applied to networks with tripartite sources, which are generalizations of the GHZ-method for the triangle and the C_4 -method for the square network.

4. In Supplementary Note there are some problems with references to Figures and Equations [between Eqs. (3) and (4) and Eqs. (40) and (41)].

5. There is redundant word 'Interestingly' in the third sentence of the section 'Certifying network links'.

6. The last sentence of the work: 'result open' → 'result opens'.

Our reply: We thank the reviewer for those last three comments, they have been corrected.